# The Generation of ROS by Exposure to Trihalomethanes Promotes the IκBα/NF-κB/p65 Complex Dissociation in Human Lung Fibroblast

**DOI:** 10.3390/biomedicines12102399

**Published:** 2024-10-20

**Authors:** Minerva Nájera-Martínez, Israel Lara-Vega, Jhonatan Avilez-Alvarado, Nataraj S. Pagadala, Ricardo Dzul-Caamal, María Lilia Domínguez-López, Jack Tuszynski, Armando Vega-López

**Affiliations:** 1Laboratorio de Toxicología Ambiental, Escuela Nacional de Ciencias Biológicas, Instituto Politécnico Nacional, Av. Wilfrido Massieu s/n, Unidad Profesional Zacatenco, Mexico City 07738, Mexico; najmtz@yahoo.com.mx (M.N.-M.); isralv@outlook.com (I.L.-V.); 2Laboratorio de Visión Artificial, Unidad Culhuacán, Escuela Superior de Ingeniería Mecánica y Eléctrica, Instituto Politécnico Nacional, Av. Santa Ana 1000, San Francisco Culhuacán CTM V, Mexico City 04440, Mexico; jonyavilez@gmail.com; 3LigronBio Inc., 10918 Technology Pl, San Diego, CA 92127, USA; nattu251@gmail.com; 4Instituto EPOMEX, Universidad Autónoma de Campeche, Av. Héroe de Nacozari No. 480, Campeche 24070, Mexico; ricadzul@uacam.mx; 5Laboratorio de Inmunoquímica I, Escuela Nacional de Ciencias Biológicas, Instituto Politécnico Nacional, Prol. Carpio y Plan de Ayala s/n, Casco de Santo Tomás, Mexico City 11340, Mexico; ldmguez@yahoo.com.mx; 6Li Ka Shing Applied Virology Institute, Department of Medical Microbiology and Immunology, University of Alberta, Edmonton, AB T6G 2E1, Canada; jack.tuszynski@gmail.com

**Keywords:** antioxidant defenses, hydrogen peroxide, IPT domain, fibroblasts, NF-κB nuclear translocation, superoxide anion

## Abstract

**Background:** Disinfection by-products used to obtain drinking water, including halomethanes (HMs) such as CH_2_Cl_2_, CHCl_3_, and BrCHCl_2_, induce cytotoxicity and hyperproliferation in human lung fibroblasts (MRC-5). Enzymes such as superoxide dismutase (SOD), catalase (CAT), and glutathione peroxidase (GPx) modulate these damages through their biotransformation processes, potentially generating toxic metabolites. However, the role of the oxidative stress response in cellular hyperproliferation, modulated by nuclear factor-kappa B (NF-κB), remains unclear. **Methods:** In this study, MRC-5 cells were treated with these compounds to evaluate reactive oxygen species (ROS) production, lipid peroxidation, phospho-NF-κB/p65 (Ser536) levels, and the activities of SOD, CAT, and GPx. Additionally, the interactions between HMs and ROS with the IκBα/NF-κB/p65 complex were analyzed using molecular docking. **Results:** Correlation analysis among biomarkers revealed positive relationships between pro-oxidant damage and antioxidant responses, particularly in cells treated with CH_2_Cl_2_ and BrCHCl_2_. Conversely, negative relationships were observed between ROS levels and NF-κB/p65 levels in cells treated with CH_2_Cl_2_ and CHCl_3_. The estimated relative free energy of binding using thermodynamic integration with the p65 subunit of NF-κB was −3.3 kcal/mol for BrCHCl_2_, −3.5 kcal/mol for both CHCl_3_ and O_2_^•^, and −3.6 kcal/mol for H_2_O_2_. **Conclusions:** Chloride and bromide atoms were found in close contact with IPT domain residues, particularly in the RHD region involved in DNA binding. Ser281 is located within this domain, facilitating the phosphorylation of this protein. Similarly, both ROS interacted with the IPT domain in the RHD region, with H_2_O_2_ forming a side-chain oxygen interaction with Leu280 adjacent to the phosphorylation site of p65. However, the negative correlation between ROS and phospho-NF-κB/p65 suggests that steric hindrance by ROS on the C-terminal domain of NF-κB/p65 may play a role in the antioxidant response.

## 1. Introduction

Under certain pathological conditions, the excessive generation of reactive oxygen species (ROS) and the resulting cellular oxidative stress are unavoidable unless counteracted by an antioxidant system [1,2]. If the cell fails to induce or regulate antioxidant protective mechanisms, pathways leading to either survival or cell death may be activated to arrest or eliminate the damaged tissue [3,4]. This raises essential questions about the molecular mechanisms involved in the shift from pro-survival signaling to pro-cell death signaling due to alterations in oxidative balance. To date, no consensus has been reached on the role of ROS in the pro-survival signaling or cell proliferation of human lung fibroblasts; however, some human fibroblasts, such as those in the skin and palmar fascia, have been shown to proliferate at low ROS concentrations. Furthermore, when ROS concentrations were inhibited in cell cultures, the proliferation of these cells was also diminished [5].

In contrast, environmental oxidants are known to cause lesions that contribute to pulmonary fibrosis by activating genes involved in cell growth, apoptosis, and fibroblast proliferation [6]. Following damage to lung tissue, fibroblasts often become excessively activated, leading to increased collagen accumulation and cell hyperproliferation as part of the repair process [7]. This excess collagen and fibroblast activity are linked to the onset of pulmonary fibrosis [8]. Beyond their direct effects on lung cells and the extracellular matrix, oxidants exacerbate pulmonary fibrosis by influencing cytokines and growth factors. In particular, the activation of nuclear factor-kappa B (NF-κB) plays a crucial role in pulmonary inflammation [9]. The NF-κB/Rel family, including RelA, RelB, NF-κB (p105/p50), NF-κB (p100/p52), and c-Rel, is involved in various immune responses, such as inflammation, immunity, differentiation, cell growth, apoptosis, and tumorigenesis [10,11,12,13]. Studies have shown that the inhibitor of kappa B kinase alpha (IκBα) forms a complex with NF-κB, with multiple residues across the protein interface affecting binding affinity. Notably, the p65-nuclear localization signal (NLS) within this complex contributes to its binding specificity [14]. Agents that activate NF-κB induce phosphorylation of IκB, leading to the release of NF-κB into the nucleus, where it regulates gene expression at κB sites [10,15,16,17]. However, the role of ROS in modulating NF-κB signaling pathways in human lung fibroblasts remains underexplored.

Prolonged or intermittent exposure to halomethanes (HMs) such as dichloromethane (CH_2_Cl_2_), trichloromethane (CHCl_3_), and bromodichloromethane (BrCHCl_2_) can lead to substantial damage to bodily tissues, particularly the liver, and kidneys, and can also result in respiratory, immunological, neurological, and developmental effects when ingested orally or inhaled. Exposure to low or chronic concentrations also leads to milder clinical respiratory symptoms such as coughing, breathlessness, or chest tightness, although no significant impact on pulmonary function has been observed. Liver and kidney damage caused by exposure to halomethanes (HMs) is primarily due to their metabolism in these organs, which are converted into reactive metabolites that generate oxidative stress. The liver plays a central role as the main organ responsible for metabolizing toxic compounds through cytochrome enzymes, particularly cytochrome P450. During this process, HMs produce reactive oxygen species ROS and other reactive intermediates, including free radicals that induce lipid peroxidation and cellular damage. In the liver, HMs are oxidized by cytochrome P450 enzymes. This process occurs after the generation of ROS, such as chloromethyl or bromomethyl radicals. [1,18,19]. These radicals attack lipids, proteins, and DNA, leading to lipid peroxidation and structural damage to hepatocytes. This oxidative stress triggers cellular necrosis or apoptosis, contributing to hepatotoxicity. The high metabolic activity of the liver makes it particularly vulnerable, as the ROS generated exceeds the organ’s antioxidant capacity [1,2,3]. For instance, CHCl_3_ produces phosgene and dichloromethyl radicals, which are responsible for lipid peroxidation and toxic effects in the liver. The kidney serves as the body’s second major organ for filtering and removing harmful substances. However, it is vulnerable to damage from reactive metabolites produced by certain substances. Oxidative stress generated by hepatic metabolism and renal excretion contributes to nephrotoxicity. Lipid peroxidation in renal cells, particularly in the proximal tubules, impairs kidney function, leading to renal failure or chronic damage. In both organs, oxidative stress caused by HMs overwhelms antioxidant defense systems, such as glutathione, exacerbating tissue damage [19].

The pulmonary tissue plays a crucial role in assessing the toxicity of harmful materials. It serves as both an entry point and an exhalation route, making it a key area for evaluation. Chronic toxicity studies are essential to identify sensitive lung targets and establish dose-response relationships in humans, particularly considering tissue-specific responses and species-related differences in HMs toxicity [18,19,20,21]. The effects in human lung fibroblasts, a different tissue type, may involve alternative pathways of oxidative stress and pro-inflammatory signaling, including activation of NF-κB.

Given that phosphorylation is crucial for activating NF-κB in response to various stresses, including oxidative and genotoxic, and that previous reports have shown that the effects of HMs are dose-dependent and vary based on the species and metabolic pathway involved, our interest in the role of ROS and oxidative stress biomarkers in NF-κB activation stemmed from a previous study where HMs induced hyperproliferation in human lung fibroblasts, leading to increased collagen synthesis [8]. While the liver and kidney are the major targets of HMs in animal studies, our research focuses on evaluating the potential impact on human lung fibroblasts, a relevant model for studying environmental exposures. These findings pointed to the activation of genes related to cell adhesion, differentiation, and proliferation through NF-κB [22]. Consequently, this study aimed to uncover the connection between oxidative stress from ROS production and NF-κB activation in human lung fibroblasts (MRC-5, ATCC cell line) exposed to chlorinated and brominated HMs, such as CH_2_Cl_2_, CHCl_3_, and BrCHCl_2_. These specific compounds were chosen because of their prevalence and documented levels in drinking water [8,23]. We also consider the toxicological profiles of these compounds, which highlight the major metabolic pathways and target organs for each halomethane, providing a broader context for the observed cellular responses. Our results suggest that the extent of chlorination in HMs significantly influences ROS induction in MRC-5 fibroblasts, mitigated primarily by the activities of CAT and GPx. Additionally, we found a negative correlation between ROS levels and phosphorylated NF-κB/p65, implying that steric hindrance by ROS at the C-terminal domain of NF-κB/p65 plays a crucial role in the antioxidant response.

## 2. Materials and Methods

### 2.1. Culture of Human Lung MRC-5 Fibroblasts and Treatments

Human lung MRC-5 fibroblasts were sourced from ATCC and cultured in Dulbecco’s Modified Eagle Medium (DMEM), as outlined in a prior study [8]. Cells were seeded at a density of 40,000 cells/mL, with 100 μL of cell suspension per well, into sterile 96-well flat-bottom plates. The culture was maintained in a 5% CO_2_ atmosphere at 37 °C, allowing cells to adhere and begin initial growth over 48 h before treatment. To optimize growth conditions and achieve 50% confluence, growth kinetics were assessed using varying cell densities (10,000 to 100,000 cells/mL) at 24, 48, and 72 h (results not shown) [8]. High-performance liquid chromatography-grade BrCHCl_2_, CHCl_3_, and CH_2_Cl_2_ were introduced to the culture medium using autoclave-sterilized 70% glycerol as the vehicle, achieving final concentrations ranging from 10^−4^ to 10^−20^ mol (nine treatments), alongside absolute control (culture medium + 10% fetal bovine serum “FBS”) and solvent control (culture medium + FBS + glycerol). Treated cells were incubated for 48 h, with media and HM concentrations refreshed at 24 h. Each treatment was replicated in three independent experiments, maintaining a vehicle concentration of 1%. Post-treatment, cells were detached using a trypsin/EDTA solution, and those from 12 wells per concentration were pooled, washed in preheated phosphate-buffered saline (PBS 1X), and centrifuged at 3200 rpm for 10 min. The resulting pellet was resuspended in 100 µL of mammalian protein extraction reagent (M-PER™) lysis solution (Thermo Scientific, Waltham, MA, USA) and sonicated for 30 s, followed by the addition of 900 µL of PBS 1X. The homogenized samples were split: one portion was centrifuged at 9000× *g* and 4 °C for 15 min to isolate the S9 fraction. Both the S9 fraction and the uncentrifuged portion were stored at −80 °C for up to two weeks until biomarker assays were conducted. The S9 fraction was designated for ROS quantification and enzymatic assays, while the uncentrifuged fraction was used for lipid peroxidation assessment.

### 2.2. Evaluation of Biomarkers

Quantification of ROS (O_2_^•^ and H_2_O_2_) and Level of Lipid Peroxidation as Thiobarbituric Acid Reactive Substances (TBARS). To measure ROS, we followed established protocols [24] using 20 μL of the S9 fraction from both treated and control cells and incorporating a final concentration of 7.0 μmol of dihydroethidium and dihydrofluorescein diacetate. The H_2_O_2_ concentration was determined through a calibration curve ranging from 0 to 7.0 mol. For O_2_^•^ quantification, we utilized a molar extinction coefficient of 4669 mol^−1^ cm^−1^, with a light path of 0.67 cm in 96-well plates containing a 200 μL final volume. Results were standardized and expressed as mol ROS per 40,000 cells. Lipid peroxidation, assessed as TBARS, was analyzed using the uncentrifuged fractions of both treated and control cells according to Buege and Aust’s method [25]. The lipid peroxidation results were expressed as TBARS, employing a molar extinction coefficient of 156,000 mol^−1^ cm^−1^, and calculated as mol of malondialdehyde (MDA) per 40,000 cells. Given that these biomarkers are independent of protein concentration, results for both ROS and lipid peroxidation were normalized to cell number.

### 2.3. Activity of the Antioxidant Defenses (SOD, CAT and GPx)

The antioxidant defense enzyme activities were measured in the S9 fraction from both treated and control cells. Superoxide dismutase (SOD; EC 1.15.1.1) activity was determined using the Misra and Fridovich method [26], employing a standard curve ranging from 3.73 to 18.65 UI of SOD derived from bovine erythrocytes. The results were reported as mmol/min/mg protein. Catalase (CAT; EC 1.11.1.6) activity was assessed following the protocol of Radi et al. [27], with the enzyme activity measured using the molar extinction coefficient of H_2_O_2_ (0.043 mmol^−1^ cm^−1^). These data were expressed as mmol/min/mg protein. Glutathione peroxidase (GPx; EC 1.11.1.9) activity was evaluated using the Lei et al. method [28]. This test was coupled with 1.0 UI of glutathione reductase, 0.03 mol of nicotinamide adenine dinucleotide (NADH), 3.5 mmol of glutathione, and 40 mmol of H_2_O_2_, with enzyme activity determined by measuring absorbance at 340 nm. The molar extinction coefficient of NADH (6.22 mmol^−1^ cm^−1^) was used for calculation, and results were reported as mmol/min/mg protein. All assays were performed in triplicate, and total protein content was quantified at 660 nm using a protein assay kit.

### 2.4. Assay of Phosphorylated NF-κB/p65 Levels

To measure NF-κB/p65 phosphorylation at Serine 536, we used the PathScan^®^ Sandwich ELISA kit from Cell Signaling Technology^®^ (Danvers, MA, USA). According to the manufacturer’s guidelines, 100 μL of the S9 fraction was applied for the assay.

### 2.5. Molecular Docking Studies

Docking studies targeting NF-κB/p65 were conducted using the MOE2019 docking module (Chemical Computing Group, Montreal, QC, Canada available at https://www.chemcomp.com/en/index.htm, accessed on 15 October 2024). The “Alpha PMI” ligand placement method was utilized, employing the London dG scoring function and the implicit generalized Born solvation model [29]. Binding sites on the NF-κB/p65 subunit (Protein Data Bank ID: 6QHL) were identified using the “alpha site finder” feature in MOE2014. A cutoff of thirty docked conformations was used by default. The binding affinities were ranked based on the ΔG scoring function (U total in kcal/mol), which integrates electrostatic and Van der Waals energies. Docking simulations employed a “reaction model” dielectric function with a cutoff of 8 to 10 Å and an interaction site radius of 6 Å. The protein–ligand complexes were then set up for molecular dynamics simulations using NAMD 2.5 software [30] and the CHARMM22 force field for proteins [31] alongside the TIP3P model for water [32]. The simulations began with a 10,000-step minimization to remove any erroneous contacts. A cutoff of 12 Å for Van der Waals interactions, with a switching function starting at 10 Å, was used. The integration time step was 2 fs, utilizing a multiple-time stepping algorithm [33,34] where covalent bond interactions were computed every step, short-range non-bonded interactions every two steps, and long-range electrostatic forces every four steps. The non-bonded interaction pair list was recalculated every 10 steps with a distance of 13.5 Å. Van der Waals and electrostatic interactions within 12 Å were considered short-range, with a smoothing function applied at 10 Å. The backbone atoms were harmonically restrained with a constant of 10.0 kcal/mol Å^2^, and the systems were heated to 300 K over 6 ps at constant volume. The simulations were equilibrated for 2 ns using the NPT ensemble (1 atm, 300 K) with the gradual removal of harmonic constraints. Following this, simulations continued for an additional 2 ns in the NPT ensemble using Langevin dynamics at 300 K with a damping coefficient of 5 ps^−1^ [35]. The pressure was maintained at 1 atm using the Langevin piston method with a piston period of 100 fs, a damping time constant of 50 fs, and a piston temperature of 300 K. Non-bonded interactions were smoothly switched off from 10 to 12 Å, with the list of non-bonded interactions truncated at 14 Å. Hydrogen bonds were kept rigid using the SHAKE algorithm, allowing a 2-fs time step. No periodic boundary conditions were applied. Atomic coordinates were saved every 1 ps during the last 2 ns of the MD simulation for trajectory analysis. CHARMM22 force field parameters were used throughout the simulations. The Root Mean Square Deviation (RMSD) was plotted against time (ns) to identify the structure with the least RMSD of the Ca trace for further analysis. Self-consistent field (SCF) energies were computed using wavefunction-based methods, constructing an initial density matrix from atomic orbital basis functions and iteratively correcting the density until self-consistency was achieved.

### 2.6. Statistical Analysis

To analyze the results, solvent controls were compared using a one-way analysis of variance (ANOVA), followed by Dunnett’s post-hoc test. For evaluating phospho-NF-κB/p65 levels, one-way ANOVA was applied, with subsequent comparison using the Kruskal–Wallis test. Relationships among biomarkers were examined through Pearson correlation analysis. Statistical significance was determined at a threshold of *p* ≤ 0.05.

## 3. Results

### 3.1. The Generation of Hydrogen Peroxide (H_2_O_2_), Superoxide Anion (O_2_^•^), and Lipid Peroxidation by Exposure to Halomethanes (CH_2_Cl_2_, CHCl_3_, BrCHCl_2_) in Human Lung Fibroblasts (MRC-5)

This study revealed that HMs were effective in inducing reactive oxygen species (ROS) in human lung fibroblasts, with varying effects based on the type and concentration of the toxicants. Specifically, CHCl_3_ led to a pronounced and statistically significant increase in ROS production across all tested concentrations, especially superoxide anions. In contrast, CH_2_Cl_2_ and BrCHCl_2_ demonstrated variable effects on ROS levels. CH_2_Cl_2_ notably increased H_2_O_2_ production in MRC-5 cells at lower concentrations (10^−20^ to 10^−14^ mol), showing a significant enhancement of 1.76- to 1.66-fold and up to a three-fold increase at higher concentrations (*p* ≤ 0.05). Conversely, BrCHCl_2_ showed an inconsistent effect on superoxide anion levels; however, at the highest tested concentrations, it caused a substantial decrease in H_2_O_2_ production, with a reduction of up to 0.17-fold (*p* ≤ 0.001) (Figure 1A,B).

### 3.2. The Activity of the Antioxidant Enzymes

No significant changes in superoxide dismutase (SOD) activity were observed in MRC-5 cells exposed to CHCl_3_. However, exposure to CH_2_Cl_2_ and BrCHCl_2_ led to irregular increases in SOD activity (Figure 3A). Catalase (CAT) activity was notably elevated across all CH_2_Cl_2_ treatments, with higher concentrations yielding a 3.48- to 3.51-fold increase (*p* ≤ 0.01). Conversely, CAT activity decreased significantly in cells exposed to BrCHCl_2_ at concentrations ranging from 10^−12^ to 10^−8^ mol (1.83- to 2.16-fold decrease; *p* ≤ 0.01), though the response varied inconsistently (Figure 3B). Glutathione peroxidase (GPx) activity increased in response to CHCl_3_ and CH_2_Cl_2_ treatments but without a clear dose-dependent trend. BrCHCl_2_ treatment, on the other hand, led to an irregular enhancement of GPx activity in MRC-5 cells (Figure 3C).

### 3.3. Evaluation of Phosphorylated NF-κB/p65 Levels

The extent of halogenation in HMs appears to play a pivotal role in NF-κB phosphorylation. In MRC-5 cells exposed to BrCHCl_2_, the phosphorylation levels of NF-κB at Ser536 exhibited a concentration-dependent increase, showing a rise from 7.15- to 13.5-fold (*p* ≤ 0.05) as concentrations escalated from 10^−14^ to 10^−6^ mol. A similar pattern was observed with CHCl_3_, where phosphorylation levels increased from 1.73- to 3.07-fold at concentrations ranging from 10^−12^ to 10^−6^ mol, though this effect was less pronounced compared with BrCHCl_2_. In contrast, exposure to CH_2_Cl_2_ did not result in any significant changes in NF-κB phosphorylation relative to control cells (Figure 4).

### 3.4. Relation Between Biomarkers

In MRC-5 cells exposed to the three halogenated compounds under study, a significant statistical relationship (*p* < 0.001) was observed between the levels of superoxide anion (O_2_^•^) and hydrogen peroxide (H_2_O_2_). Lipid peroxidation, as indicated by TBARS, correlated with ROS levels in cells treated with CH_2_Cl_2_ and BrCHCl_2_. However, for CHCl_3_, only a significant association between TBARS and H_2_O_2_ was found. Superoxide dismutase (SOD) activity did not correlate with ROS levels, except in the case of BrCHCl_2_. Conversely, catalase (CAT) activity was associated with H_2_O_2_ across all treatments and showed correlations with O_2_^•^ and SOD in cells treated with CH_2_Cl_2_ and BrCHCl_2_. CAT activity also exhibited a statistical relationship with TBARS in cells exposed to chlorinated HMs. Glutathione peroxidase (GPx) activity correlated with H_2_O_2_ in cells exposed to BrCHCl_2_ and with TBARS in cells treated with CHCl_3_. Notably, both ROS levels were inversely related to phosphorylated NF-κB/p65 protein at Ser536 in cells treated with chlorinated HMs, with O_2_^•^ also showing a link to this nuclear factor. On the other hand, phosphorylated NF-κB/p65 protein at Ser536 exhibited an inverse correlation with TBARS in CH_2_Cl_2_-treated cells and with CAT activity in cells exposed to BrCHCl_2_ (Table 1).

### 3.5. Molecular Docking Analysis

In this study, MOE simulations were run until multiple docked conformations were achieved, with the best solution for the ligand/receptor model being determined based on the predicted energy analysis and result consistency. The in-silico findings revealed that the estimated free energy of binding to the p65 subunit of NF-κB was −7.0 kcal/mol for both BrCHCl_2_ and H_2_O_2_, −6.5 kcal/mol for CHCl_3_, and −7.6 kcal/mol for O_2_^•^. These interaction energy values suggest that these ligands bind effectively to the nuclear factor. For CHCl_3_, Cl1 was found to interact with Thr191 at 3.94 Å, while Cl2 bound with Thr191, Ala192, Leu194, and Ser281 at distances of 3.37, 2.96, 3.79, and 3.60 Å, respectively. Cl3 was also in contact with Leu194 at 3.76 Å within the RHD region of the IPT domain (Figure 5A). In BrCHCl_2_, Cl1 was closely associated with Glu193 and Leu194, interacting with backbone nitrogen atoms at distances of 2.9 and 2.83 Å, respectively, with an electrostatic energy of 0.9 kcal/mol (Figure 5B). Cl2 and Br ions interacted with the backbone carbon and side chain oxygen of Thr191 at a distance of 3.0 Å. This indicates that backbone nitrogen plays a critical role in the interactions of Cl1 and Cl2 with the RHD region of the IPT domain of p65 of NF-κB. The biotransformation of CHCl_3_ and BrCHCl_2_ produced ROS (O_2_^•^ and H_2_O_2_), suggesting that both chemical species interact with the NF-κB/p65 complex. O_2_^•^ formed four hydrogen bonds with backbone nitrogens and carbon atoms of Thr191, Ala192, Glu193, and Leu194, while H_2_O_2_ formed five hydrogen bonds with these same residues and an additional bond with the side chain oxygen of Leu280 (Figure 6A,B). The RMSD analysis of Ca trace against time (ns) showed that H_2_O_2_ exhibited higher stability compared with other ionic species, with the order being CHCl_3_ < O_2_^•^ < BrCHCl_2_ < H_2_O_2_ (Figure 7A). Moreover, the NF-κB/p65-H_2_O_2_ complex formed more hydrogen bonds than the other three complexes, indicating greater stability (Figure 7B). Additionally, the calculated HUMO-LUMO energies for the NF-κB/p65-ligand complex showed that H_2_O_2_ and O_2_^•^ were chemically stable, with values of −0.3 and −0.5 kcal/mol, respectively. This correlates well with ΔG values from thermodynamic integration and RMSD stability from MD simulations. SCF calculations indicated that CHCl_3_ has a higher affinity with −46.4 a.u. and a ΔH of −23.6 kcal/mol, as detailed in Table 2.

## 4. Discussion

Reactive oxygen species (ROS) have a significant impact on lung tissue, with nearly all lung cell types capable of producing these highly reactive molecules. Under normal conditions, ROS are generated by NADPH oxidase and serve as secondary messengers, activating various intracellular proteins and enzymes involved in both physiological and pathological processes [23,36]. Pulmonary fibroblasts, in particular, generate ROS, especially when stimulated by inflammatory cytokines [37]. However, excessive ROS production can lead to DNA damage, cellular morphological changes, and lung injury if the antioxidant defense system is overwhelmed [38,39]. Additionally, environmental toxicants can exacerbate ROS generation [40,41]. There is limited information regarding ROS production in human lung fibroblasts induced by HMs, by-products of water disinfection processes. Exposure to HMs occurs through ingestion, inhalation, and dermal contact, with these compounds being readily absorbed via the skin, respiratory tract, or gastrointestinal system and subsequently accumulating in the stomach, liver, and kidneys. Primarily, unmetabolized HMs are expelled through exhalation. For example, trichloromethane (CHCl_3_) undergoes extensive conjugation with dose-dependent metabolism: about 50% of an oral dose is converted to CO_2_, 38% is metabolized by the liver, and less than 17% is exhaled unchanged [19]. The inhalation of HMs poses a direct threat to lung tissue, highlighting the crucial role of pulmonary surfactant (PS). PS, composed of lipids and proteins, is essential for innate immunity and efficient gas exchange [42,43]. Acute lung injury can disrupt PS function through various inhibitors, including ROS and reactive nitrogen species, impairing surfactant activity [42,44]. This disruption facilitates HMs penetration into lung cells, such as pulmonary fibroblasts. Passive exposure to HMs indoors also represents a potentially underestimated source of PS dysfunction [45]. Thus, examining the cumulative effects of HMs on lung tissue and the efficacy of antioxidant defenses, especially PS, is crucial.

The concentrations studied (10^−4^ to 10^−20^ mol) reflect biologically relevant levels reported in previous research. For instance, cancer risk thresholds for HMs have been identified at 3.36 × 10^−7^ mol for females and 2.44 × 10^−7^ mol for males calculated with BrCHCl_2_ (55 μg/L and 40 μg/L, respectively), and a dose-dependent effect for brominated HMs above 2.1 × 10^−7^ mol BrCHCl_2_ (35 μg/L) has been noted in recent meta-analyses of disinfection by-products in municipal water supplies [46]. Additionally, the US EPA has set a maximum contaminant level for HMs at 80 μg/L (4.88 × 10^−7^ mol calculated for BrCHCl_2_) to mitigate long-term health risks [47]. In air studies, HM concentrations after disinfection ranged from 8.84 × 10^−9^ to 2.56 × 10^−8^ mol/m^3^ for BrCHCl_2_ (1.46–4.20 μg/m^3^), with CHCl_3_ detected in 10 of 11 locations between 1.0 × 10^−8^ and 3.0 × 10^−8^ mol/m^3^ (1.20–3.59 μg/m^3^) [45,48]. The present study reveals that the toxicants examined can induce reactive oxygen species (ROS) in vitro, with a notable emphasis on superoxide anion (O_2_^•^). This finding aligns with similar observations in peripheral blood mononuclear cells (PBMCs) from *Cyprinus carpio carpio* treated with dichloromethane (CH_2_Cl_2_), trichloromethane (CHCl_3_), and bromodichloromethane (BrCHCl_3_) [23]. Although direct data on ROS production in human fibroblasts exposed to HMs are sparse, it is proposed that ROS generation involves multiple, possibly non-sequential events and their interplay with pro-oxidant forces and oxidative damage, outlined as follows: (1) Biotransformation processes play a crucial role in ROS production. (2) ROS generation is not solely governed by mitochondrial activity. (3) The likelihood of ROS generation increases with the number of halogens present. (4) Physicochemical properties, such as the electronegativity of metabolites formed during biotransformation, influence ROS production rates. (5) The induction of ROS is largely dependent on both specific and non-specific antioxidant defenses. These data suggest that ROS generation might occur through pathways independent of mitochondrial complex I activity [23], such as through electron flow disruption during cytochrome P450 isoenzyme redox processes (e.g., CYP 2E1). Other enzymes, including glutathione S-transferase isoform tetha (GSTT), are also involved in the bioactivation of HMs [8,23,49,50,51]. The results indicate that the chlorination degree of HMs is a significant factor in ROS induction in MRC-5 cells. Initially, all three HMs studied can induce comparable levels of O_2_^•^ mediated by CYP 2E1 [23]. Subsequently, HMs with three halogens undergo reductions, dehalogenations, and reductive dehalogenations mediated by enzymes such as GSTT, further promoting ROS generation [8,52]. However, CHCl_3_ generated higher levels of O_2_^•^ compared with BrCHCl_2_, likely because of the release of hydrochloric acid (HCl) with a higher electronegativity (δ = 1.06) than hydrobromic acid (HBr) released from BrCHCl_2_ (δ = 0.86) during dehalogenation [23]. Conversely, CH_2_Cl_2_ produced the lowest levels of O_2_^•^, potentially due to fewer reductive dehalogenations during its biotransformation [23]. Furthermore, the final biotransformation step of CHCl_3_ mediated by GSTT likely increases respiratory rate and proton motive force due to H^+^ release, thus facilitating ROS generation through mitochondrial complex I and the electron transport chain [23,53]. On the other hand, the significant reduction in O_2_^•^ levels in MRC-5 fibroblasts exposed to high concentrations of CH_2_Cl_2_ can be attributed to enzymatic dismutation of H_2_O_2_ by superoxide dismutase (SOD), which is crucial for maintaining antioxidant defense in the lungs [6,54,55]. SOD converts O_2_^•^ into H_2_O_2_, thereby preventing further chain reactions that could generate hydroxyl (^•^OH) and peroxynitrite radicals (ONOO^−^). Nevertheless, O_2_^•^ may convert to ^•^OH in the presence of metal ions via Fenton and Haber–Weiss reactions [56]. Excessive ROS production in the lungs disrupts the balance between oxidants and antioxidants, contributing to airway pathologies [57,58,59,60]. Elevated ROS levels can also damage critical cellular components, including proteins, unsaturated lipids, and DNA, thereby compromising cellular homeostasis [56,61]. Conversely, the generation of H_2_O_2_ from the dismutation of O_2_^•^ can occur both spontaneously (without a catalyst) and through the action of superoxide dismutase (SOD) [56]. Notably, in MRC-5 cells treated with CH_2_Cl_2_ at concentrations of 10^−20^ mol, the highest H_2_O_2_ levels did not coincide with the peak concentration of its precursor (O_2_^•^). This observation supports the idea of non-catalyzed dismutation occurring early in the experiment rather than relying on SOD activity, as previously documented [56]. In the present study, while H_2_O_2_ levels in MRC-5 cells exposed to CHCl_3_ were correlated with O_2_^•^ levels, this relationship did not extend to SOD activity. This suggests that in the case of CH_2_Cl_2_ treatment, non-catalyzed dismutation of O_2_^•^ to H_2_O_2_ may have been facilitated by the H+ ions generated during late-stage biotransformation [23]. When H_2_O_2_ remains stable in the cell’s internal environment and escapes breakdown by catalase (CAT) or glutathione peroxidase (GPx), it can activate various signaling pathways, influencing processes such as cell proliferation, differentiation, migration, or apoptosis [23,62,63,64,65,66]. In contrast, for MRC-5 fibroblasts treated with CH_2_Cl_2_ and BrCHCl_2_, H_2_O_2_ levels showed a statistically significant correlation with O_2_^•^ concentrations and exhibited a similar response at lower doses. However, in cells exposed to BrCHCl_2_, the decrease in H_2_O_2_ was associated with the activity of CAT and GPx. While CAT is crucial for the dismutation of H_2_O_2_, GPx plays a significant role in maintaining the redox balance in lung cells exposed to BrCHCl_2_. This enzyme not only removes organic hydroperoxides but also contributes to the glutathione redox cycle, which is prevalent in alveolar epithelial lining fluid [67]. Despite the involvement of various antioxidant enzymes, ROS induction led to lipid peroxidation in vitro in human lung fibroblasts.

Building on the previously documented cytotoxic effects of HMs in MRC-5 cells [8], this study further reveals that CHCl_3_ induces significant lipid peroxidation, as assessed by TBARS, particularly at higher concentrations. This lipid peroxidation is directly related to the concentration of HMs, with the damage primarily stemming from ROS that remains unneutralized by the antioxidant defense system. This imbalance between pro-oxidant forces (ROS) and antioxidants (SOD, CAT, and GPx) initiates a cascade of radical reactions: (1) initiation driven by the dominance of pro-oxidants, (2) propagation of free radicals causing extensive damage to the lipid bilayer, and (3) potential polymerization of free radicals. Lipid peroxidation is a damaging oxidative process involving the deterioration of the lipid bilayer, initially mediated by hydroxyl radicals (^•^OH), which are the most significant oxidizing species in this process [68]. Chain reactions may also produce alkyl hydroperoxides (LOOH), alkyl peroxyl radicals (LOO^•^), and alkoxyl radicals (LO) [56]. The extent of lipid peroxidation induced by CHCl_3_ correlates with H_2_O_2_ levels, suggesting significant damage to lung fibroblast membranes, which could disrupt lipid bilayer fluidity and functionality, potentially leading to extensive alterations in hormone receptors and signaling proteins [56]. Previous studies have shown that high concentrations of CHCl_3_ reduce MRC-5 cell growth and induce cytotoxicity [8]. This suggests that lipid peroxidation caused by CHCl_3_ may contribute to reduced cell growth and increased cytotoxicity. However, the lungs have mechanisms to counteract such damage. Kornbrust and Mavis [69] observed that lipid peroxidation is substantially higher in highly oxygenated tissues such as the lungs and heart compared with other organs such as the liver and kidneys. Lung fibroblasts are particularly adept at remodeling and repairing damaged tissues, with the lung demonstrating efficient regeneration through progenitor cell activation and cell proliferation [70,71,72]. Interestingly, exposure to higher concentrations of CHCl_3_ led to a decrease in TBARS levels in MRC-5 cells at 10^−6^ mol, potentially due to increased GPx activity. Conversely, a significant inverse relationship between TBARS and CAT activity was noted, suggesting that oxidative damage from H_2_O_2_ was reduced as CAT activity increased, which correlated with a decrease in lipid peroxidation. In contrast, BrCHCl_2_ exposure resulted in increased lipid peroxidation at median concentrations, which was associated with ROS levels. However, higher concentrations of BrCHCl_2_ significantly reduced oxidative damage, possibly because of increased GPx activity, despite a lack of significant correlation. Given the lung’s capacity for cell repair and regeneration following oxidative injury, examining relationships with other biomarkers involved in cell cycle regulation, such as NF-κB, could provide further insights.

Interestingly, only halomethanes with three halogens, specifically BrCl_2_CH and Cl_3_CH, were observed to elevate levels of phospho-NF-κB/p65 protein at Ser536. The extent of this increase and its correlation with oxidative stress biomarkers varied. While quantitative structure-activity relationships (QSARs) could partially explain the differential responses in NF-κB translocation to the nucleus, molecular docking studies offer a more detailed understanding. These studies reveal how HM treatment disrupts the IκBα-NF-κB/p65 complex, providing insights into the molecular mechanisms underlying the observed effects.

The in-silico analysis using Molecular Operating Environment (MOE) software (MOE2019 docking module available at https://www.chemcomp.com/en/index.htm; accessed on 15 October 2024) revealed that BrCHCl_2_ and Cl_3_CH exhibit estimated free binding energies of −7.0 kcal/mol and −6.5 kcal/mol, respectively, when interacting with the p65 subunit of NF-κB. These interaction energy values suggest that these ligands form robust bonds with the nuclear factor, correlating with increased phosphorylation levels of NF-κB/p65 in a concentration-dependent manner, particularly with BrCHCl_2_. Notably, the chloride and bromide groups from these halomethanes interact within the C-terminal domain of NF-κB/p65, an area critical for gene expression regulation because of its transcriptional activation domain (TAD) [73]. Our findings highlight that Cl1, Cl2, and bromide interact with four adjacent amino acid residues (Thr191, Ala192, Glu193, and Leu194) in the Rel Homology Domain (RHD), which is essential for DNA binding. The IPT domain, spanning amino acids from Thr191 to Asp291, is responsible for the dimerization of NF-κB/p65 and related transcription factors [74,75]. This domain is crucial for regulating responses to various stress conditions, including oxidative stress and chemical exposure [76,77,78]. Additionally, conserved residues in the dimer interface, such as Arg198, Glu211, Leu215, and Cys216, play a significant role in hydrophobic interactions, aligning closely with the binding sites of Cl1, Cl2, and bromide anions [79].

Conversely, post-translational modifications, including phosphorylation, acetylation, and methylation, can significantly influence the transcriptional activity of NF-κB [80]. Notably, the phosphorylation of IκB proteins triggers their ubiquitination and subsequent degradation by proteasomes, which releases the NF-κB/p65 homodimer. This release enables p65 to bind DNA either as a homodimer or in combination with other subunits [81,82,83]. Several phosphorylation sites within the IPT domain, near the binding residues of Cl1, Cl2, and bromide anions from the halomethanes, are particularly noteworthy. For instance, residues 254, 276, and 281, with a particular emphasis on Ser281, are crucial. Previous site-directed mutagenesis studies have identified S205, S276, and S281 as essential for the transcriptional activity of p65 [17,84]. These findings, combined with our results, suggest that the phosphorylation of NF-κB subunits significantly impacts its function. Specific phosphorylation events are involved in the selective regulation of NF-κB transcriptional activity in a gene-specific context [85]. Consistent with our results, the interaction of the backbone carbon of CHCl_3_ with Ser281 may alter the conformation of the IκBα/p65 complex, contributing to the hyperproliferation of MRC-5 cells, as noted in previous research [8]. It has been observed that phosphorylation of p65 induces conformational changes that affect its ubiquitination, stability, and protein–protein interactions. NF-κB/p65 is a highly dynamic protein, not rigidly fixed but capable of undergoing structural changes following activation [86]. We hypothesize that the interaction of Cl1, Cl2, and bromide from BrCHCl_2_ and CHCl_3_ may disrupt the p65 homodimer’s structure and its interaction with IκBα, leading to the dissociation of the IκBα-NF-κB complex and activation of target gene transcription. However, to fully understand these effects in human fibroblasts, it is essential to consider the impact of reactive metabolites generated during biotransformation, such as ROS.

Although previous research has explored the relationships between oxidative stress responses and NF-κB/p65 activation [87,88,89,90,91,92,93,94,95,96,97], there is a notable absence of data regarding ROS production in human lung fibroblasts induced by HMs. Prior studies have documented ROS induction in peripheral blood mononuclear cells (PBMCs) from *Cyprinus carpio carpio* treated with CH_2_Cl_2_, CHCl_3_, and BrCHCl_2_ [23]. In the present study, we observed that these toxicants induce ROS production in vitro, with binding energies estimated at −7.0 kcal/mol for H_2_O_2_ and −7.6 kcal/mol for O_2_^•^ in relation to NF-κB/p65. These ROS interactions primarily affect the IPT domain of NF-κB/p65, suggesting their role in activating target genes associated with antioxidant responses, as indicated by Pearson correlation analysis. NF-κB-regulated genes are crucial for modulating ROS levels within the cell, given that ROS can either inhibit or stimulate NF-κB signaling pathways. Specifically, interactions with cysteine residues are vital for disrupting NF-κB pathways; for instance, hydrogen peroxide can inhibit IKK activation by targeting these cysteine residues, thereby impacting tyrosine phosphatases [98,99]. Our biochemical analysis of MRC-5 cells treated with HMs, complemented by computational studies using MOE2019, revealed that BrCHCl_2_ and CHCl_3_ promote NF-κB/p65 phosphorylation by activating the IPT domain. This activation is likely linked to the observed hyperproliferation of these cells, as documented previously [8]. However, ROS generated during the metabolism of these toxicants appear to disrupt IPT domain activation involved in the antioxidant response mediated by NF-κB/p65. This disruption is evident from the negative correlations between ROS and the antioxidant enzymes SOD, CAT, and GPx in cells treated with CHCl_3_.

On the other hand, cell lines, such as MRC-5, provide a consistent and reproducible platform for experiments, dissimilar to primary cells, which can vary between isolations. MRC-5 cells, derived from human lung fibroblasts, are normal diploid cells with a limited lifespan before senescence, making them a valuable model for studying biological processes and diseases affecting human lung tissue [100,101]. These cells express critical antioxidant enzymes such as SOD, catalase, and glutathione peroxidase, similar to those found in native pulmonary fibroblasts. These enzymes are essential for detoxifying ROS and protecting against oxidative stress. MRC-5 cells also synthesize glutathione, a key indicator of cellular antioxidant capacity [102], and can activate signaling pathways in response to oxidative stress, including transcription factors such as Nrf2, which regulate antioxidant and detoxification gene expression [103]. Despite these similarities, MRC-5 cells, as an established cell line, may not fully replicate the complex microenvironment and cellular interactions present in in situ lung tissue. Factors such as the extracellular matrix, immune cells, and paracrine signals significantly influence the antioxidant response in native tissue [104]. Thus, further research using more complex systems, such as in vivo models, is necessary to understand these responses and their potential implications better.

## 5. Conclusions

The level of chlorination in CHCl_3_ and BrCHCl_2_ plays a crucial role in triggering the production of reactive oxygen species (ROS) in MRC-5 human lung fibroblasts. The increased activity of antioxidant enzymes, such as catalase and glutathione peroxidase, can partially alleviate this oxidative stress. As these trihalomethanes undergo biotransformation, the formation of harmful metabolites such as H_2_O_2_ and O_2_^•−^ can impact the phosphorylation sites on the IPT domain of the NF-κB/p65 complex. Our research suggests that BrCHCl_2_ has a stronger binding affinity to the NF-κB/p65 complex, leading to a significant increase in phospho-NF-κB/p65 levels. This indicates that BrCHCl_2_ may more effectively facilitate the dissociation of the IκBα-NF-κB complex, thereby enhancing the transcription of pro-inflammatory target genes. Furthermore, ROS may disrupt the expression of other genes, including those responsible for antioxidant enzymes, potentially compromising the cell’s ability to combat oxidative damage.

## Figures and Tables

**Figure 1 biomedicines-12-02399-f001:**
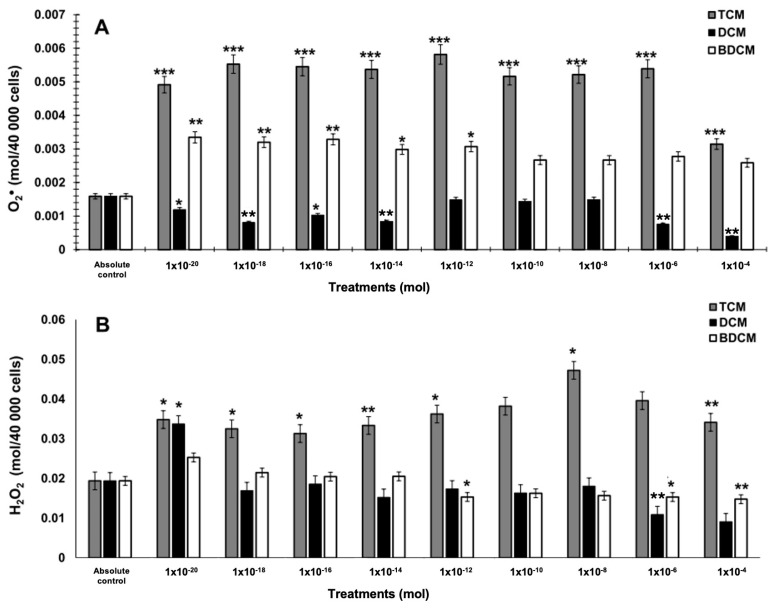
Levels of superoxide anion (O_2_^•^) and hydrogen peroxide (H_2_O_2_) in human lung fibroblasts (MRC-5) exposed to dichloromethane (CH_2_Cl_2_), trichloromethane (CHCl_3_), and bromodichloromethane (BrCHCl_2_). Panel (**A**) displays the measurements of superoxide anion, while Panel (**B**) shows hydrogen peroxide levels. Statistical significance compared with the absolute control is denoted by * *p* ≤ 0.05, ** *p* ≤ 0.01, and *** *p* ≤ 0.001. The abbreviations used are DCM for CH_2_Cl_2_, TCM for CHCl_3_, and BDCM for BrCHCl_2_. Exposure to elevated concentrations of CHCl_3_ (ranging from 10^−10^ to 10^−6^ mol) significantly increased lipid peroxidation, with levels rising between 10- and 15.6-fold. BrCHCl_2_ also heightened lipid peroxidation at 10^−10^ mol, with a 1.87-fold increase. However, higher concentrations of BrCHCl_2_ (10^−8^ and 10^−6^ mol) caused a notable decrease in lipid peroxidation, reducing TBARS levels by approximately 0.12- to 0.78-fold. In contrast, exposure to CH_2_Cl_2_ did not produce significant changes in lipid peroxidation, although a slight reduction in oxidative damage was observed at 10^−6^ mol, with a decrease of around 0.15-fold in MRC-5 cells (Figure 2).

**Figure 2 biomedicines-12-02399-f002:**
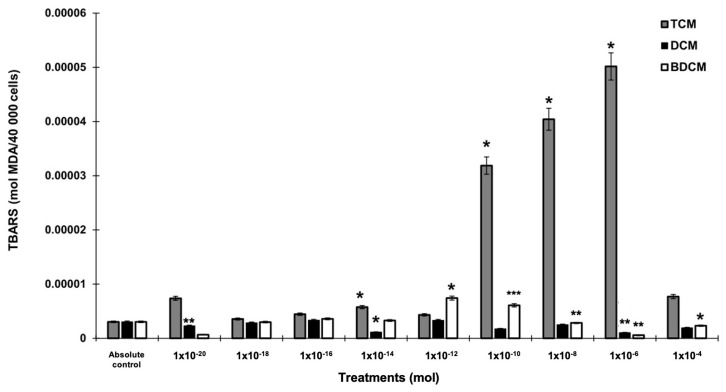
Lipid peroxidation levels, measured as thiobarbituric acid reactive substances (TBARS), in human lung fibroblasts (MRC-5) exposed to dichloromethane (CH_2_Cl_2_), trichloromethane (CHCl_3_), and bromodichloromethane (BrCHCl_2_). Statistical significance compared with the absolute control is indicated by * *p* ≤ 0.05, ** *p* ≤ 0.01, and *** *p* ≤ 0.001. Abbreviations: DCM = CH_2_Cl_2_; TCM = CHCl_3_; BDCM = BrCHCl_2_.

**Figure 3 biomedicines-12-02399-f003:**
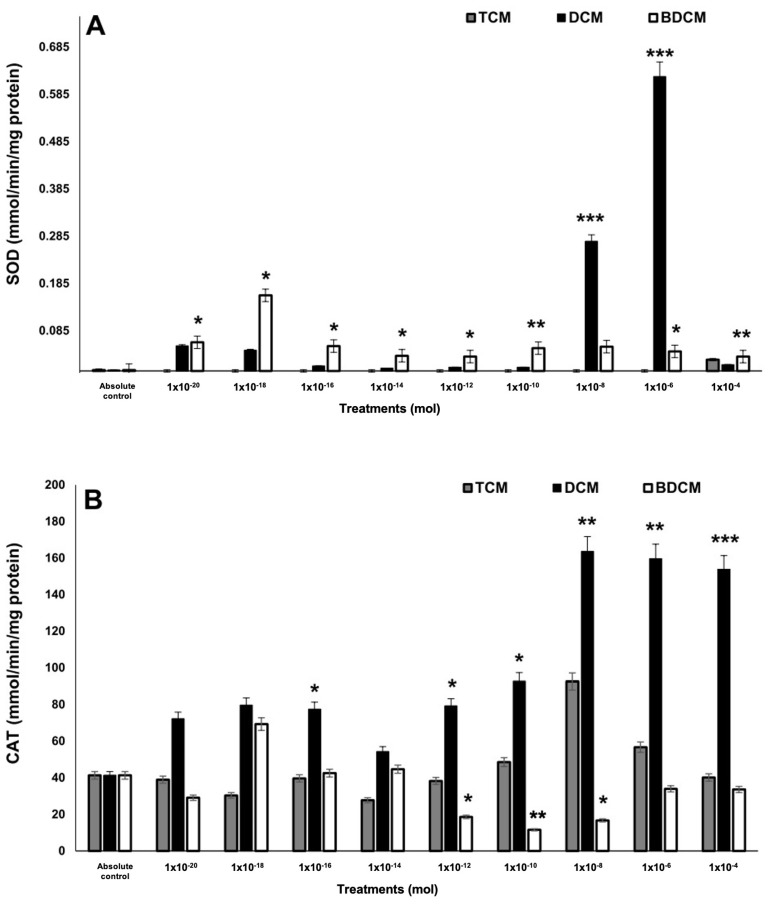
The activity of the enzymes involved in the antioxidant response on human lung fibroblasts (MRC-5) exposed to dichloromethane (CH_2_Cl_2_), trichloromethane (CHCl_3_), and bromodichloromethane (BrCHCl_2_). (A) Superoxide dismutase. (B) Catalase. (C) Glutathione peroxidase. Statistical differences regarding absolute control with * *p* ≤ 0.05, ** *p* ≤ 0.01, and *** *p* ≤ 0.001. Abbreviations: DCM = CH_2_Cl_2_; TCM = CHCl_3_; BDCM = BrCHCl_2_.

**Figure 4 biomedicines-12-02399-f004:**
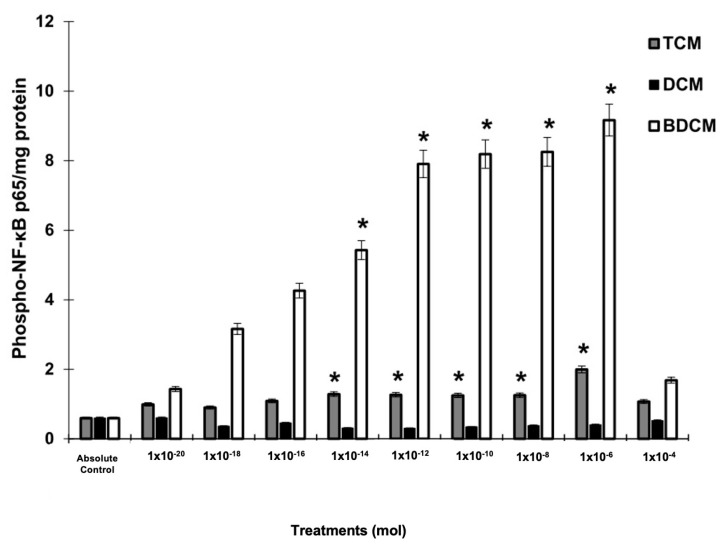
Phosphorylation levels of NF-κB/p65 at Ser536 in human lung fibroblasts (MRC-5) following exposure to dichloromethane (CH_2_Cl_2_), trichloromethane (CHCl_3_), and bromodichloromethane (BrCHCl_2_). Statistical significance relative to the absolute control is indicated with * *p* ≤ 0.05. Abbreviations: DCM = CH_2_Cl_2_; TCM = CHCl_3_; BDCM = BrCHCl_2_.

**Figure 5 biomedicines-12-02399-f005:**
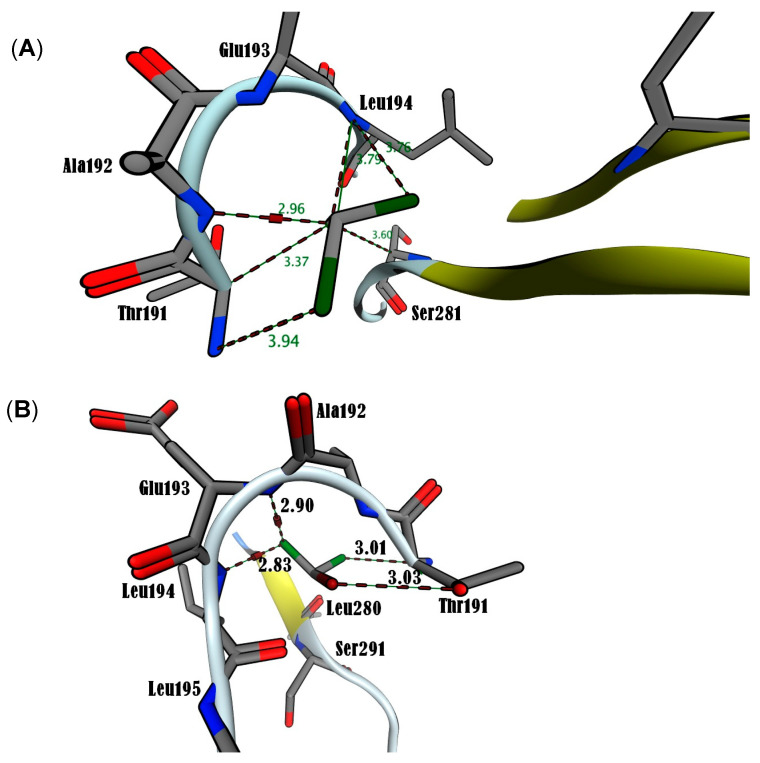
Molecular docking analysis using MOE2019 illustrates the interactions of trihalomethanes with the p65 subunit of NF-κB. (**A**) Interaction of trichloromethane (CHCl_3_) with p65. (**B**) Interaction of bromodichloromethane (BrCHCl_2_) with p65. C bonds (grey), Cl atoms (green), Br atoms (brown), O atoms (red), and N atoms (blue). Distances are shown in Å.

**Figure 6 biomedicines-12-02399-f006:**
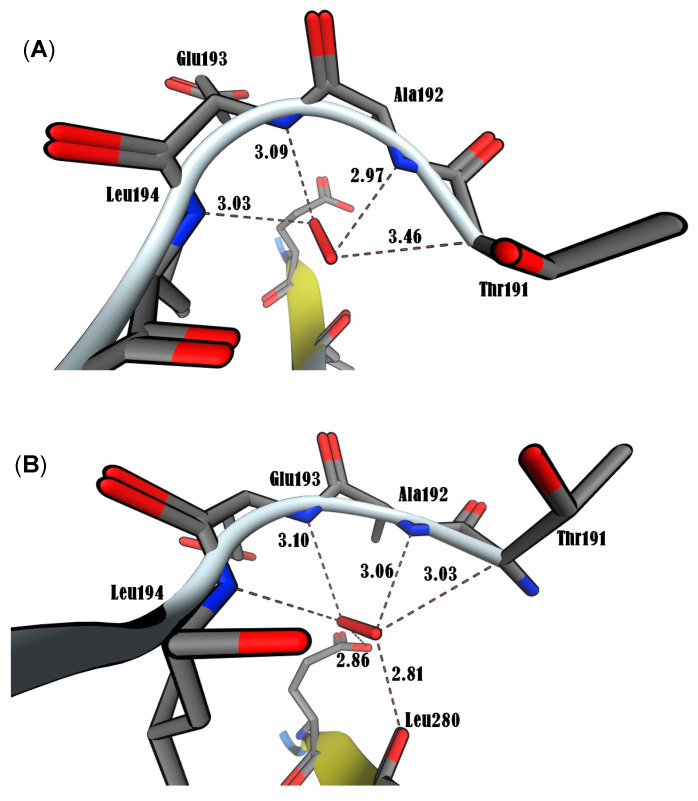
Molecular docking analysis that shows the interaction of ROS with p65 of NF-κB using MOE2019. (A) Superoxide anion (O_2_^•^). (B) hydrogen peroxide (H_2_O_2_). C bonds (grey), O atoms (red), and N atoms (blue). Distances are shown in Å.

**Figure 7 biomedicines-12-02399-f007:**
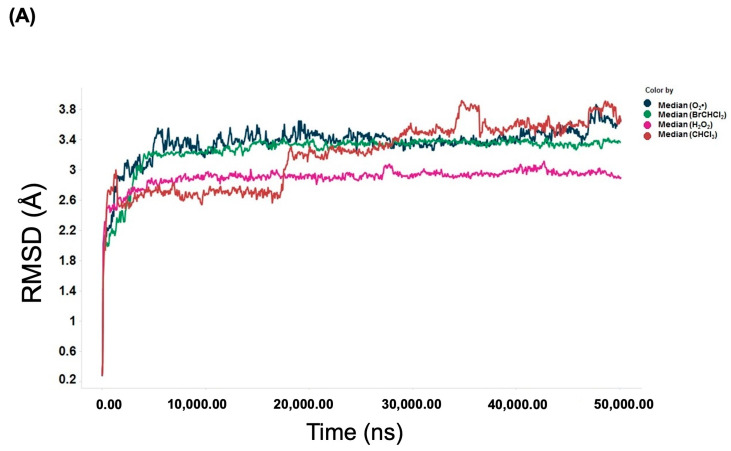
Molecular dynamics (MD) trajectory analysis showing the RMSD of Ca traces plotted against time (ns) and the number of hydrogen bonds formed by bromodichloromethane (BrCHCl_2_), trichloromethane (CHCl_3_), superoxide anion (O_2_^•^), and hydrogen peroxide (H_2_O_2_) with NF-κB/p65. (**A**) RMSD values. (**B**) Number of hydrogen bonds.

**Table 1 biomedicines-12-02399-t001:** Correlation of Biomarkers in Human Lung Fibroblasts (MRC-5) Exposed to Dichloromethane (CH_2_Cl_2_), Trichloromethane (CHCl_3_), and Bromodichloromethane (BrCHCl_2_). This table displays the correlation coefficient (R) and the associated confidence level (*p*) for significant relationships identified through Pearson correlation analysis. Positive R values indicate direct relationships between pairs of biomarkers, while negative R values denote inverse relationships. Only statistically significant results for each halogenated compound are presented.

	R, *p*	R, *p*	R, *p*	R, *p*	R, *p*	R, *p*
	CH_2_Cl_2_					
	H_2_O_2_	TBARS	SOD	CAT	GPx	NF-κB
O_2_^•^	0.590, <0.001	0.728, <0.001		0.596, <0.001		−0.533, <0.01
H_2_O_2_		0.736, <0.001		0.541, <0.01		
TBARS				0.408, <0.05		−0.471, <0.01
SOD				0.584, <0.001		
	CHCl_3_					
	H_2_O_2_	TBARS	SOD	CAT	GPx	NF-κB
O_2_^•^	0.884, <0.001				0.504, <0.01	−0.611, <0.001
H_2_O_2_		0.465, <0.05		0.592, <0.001		−0.672, <0.001
TBARS				0.771, <0.001	0.514, <0.01	
	BrCHCl_2_					
	H_2_O_2_	TBARS	SOD	CAT	GPx	NF-κB
O_2_^•^	0.915, <0.001	0.418, <0.05	0.539, <0.01	0.682, <0.001	0.407, <0.05	
H_2_O_2_		0.404, <0.05	0.396, <0.05	0.653, <0.001	0.451, <0.05	
SOD				0.536, <0.01		
CAT						−0.406, <0.05

O_2_^•^, Superoxide anion; H_2_O_2_, Hydrogen peroxide; TBARS, lipid peroxidation as thiobarbituric reactive substances; SOD, the activity of superoxide dismutase; CAT, the activity of catalase; GPx, the activity of glutathione peroxidase; NF-κB, levels of the nuclear factor-kappa B.

**Table 2 biomedicines-12-02399-t002:** Self-consistent field (SCF) energies for BrCHCl_2_, CHCl_3_, H_2_O_2_, and O_2_^•^ in complex with NF-κB/p65, calculated using wavefunction-based methods. The affinity scoring function, ΔG (U total in kcal/mol), was used to evaluate and rank the candidate poses based on the combined electrostatic and Van der Waals energies.

Compound	SCF (a.u.)	ΔH (kcal/mol)	ΔE(eV)	ΔG (kcal/mol)
CHCl_3_	−46.4	−23.6	11.6	−3.5
BrCHCl_2_	−23.9	−17.9	12.5	−3.3
O_2_^•^	−24.4	−35.3	0.5	−3.6
H_2_O_2_	−45	−15.5	0.3	−3.6

## Data Availability

Data supporting reported results are available under request.

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
