# Peer review of "The Generation of ROS by Exposure to Trihalomethanes Promotes the IκBα/NF-κB/p65 Complex Dissociation in Human Lung Fibroblast"

_biomedicines, 2024, doi:10.3390/biomedicines12102399_

Round 1
Reviewer 1 Report
Comments and Suggestions for Authors
The authors have incubated cultures of lung fibroblasts with different concentrations of the trihalomethanes BrCHCl2, CHCl3, and CH2Cl2 , being byproducts of drinking water disinfection to determine ROS and reactive oxygen species a parameters for oxidative stress and parameters indicative for antioxidative defense mechanisms.
All this is acceptable if the biological relevance of the findings is discussed and presented. Human exposure to the trihalomethanes may occur via skin, the gut and the lung. All these tissues contain antioxidative systems to scavenge reactive oxygen species, which are constantly generated. These systems need to be overwhelmed before detrimental cellular effects occur. It needs to be demonstrated whether the effective concentrations observed in the study occur in vivo. This needs to consider the role of surfactants in the lung, the protective sweat and cellular layers on the skin and the even more complex situation in the gastrointestinal tract.
Moreover, the “human” fibroblast cells are a permanent cell-line. It needs to be defined whether the antioxidative system of these cells resembles that of the lung fibroblasts in situ.
Without this additional information the data are of little added value: A cell line has been incubated with certain chemicals and the response is measured. Whatever this means remains obscure.
Comments on the Quality of English LanguageNo further comments.
Reviewer 2 Report
Comments and Suggestions for Authors
This manuscript described that the degree of chlorination of HMs may be a key factor in the induction of ROS in MRC-5 lung human fibroblasts and counter this damage mainly through the activity of CAT and GPx. Furthermore, the authors showed a negative correlation between ROS and phosphorylated NF-κB/p65, suggesting that steric hindrance of ROS at the C-terminal domain of NF-κB/p65 is involved in the antioxidant response. Although the results shown were interesting, the explanation for some of the contradictory results seemed insufficient. The authors should address against some points shown as follows;
1. To give a brief overview of what the authors show, English might be edited with a help of a native English-speaking scientist or a commercially available English proofreader.
2. What did "enzymes" refer to in the second line of the “Abstract”?
3. “HM” may mean “halomethanes”, so please specify the abbreviation as halomethanes (HM) when “HM” was first appearance. Once the authors specify the abbreviation, please continue to use it until the end (for example, at the end of line 356).
4. Why were the three halomethanes shown in this manuscript selected from the many available halomethanes? Please briefly explain the reason in the "Introduction".
5. In Section 2.4, please clearly indicate the product name and manufacturer of the kit used.
6. In Figures 1A, 1B and 2, why did the amount of superoxide anion, hydrogen peroxide and TBARS decrease significantly in the highest concentration TCM exposure group, respectively? What mechanisms were thought to be involved in this decrease?
7. From the results in Figure 3C, it can be considered that GPx was low only in the group exposed to 0.001 mol of HMs?
8. In Figure 2, what do the asterisks shown in Figure 2 indicate significant differences? What did the different colors of the asterisk represent?
9. In Figure 4, although the authors treated the TCM-exposed groups and the BDCM-exposed groups equally because they had a significant increase of phosphorylated NFkB-p62, the difference in the amounts of phosphorylated NFkB-p62 between the two exposure groups cannot be ignored.
Comments on the Quality of English LanguageTo give a brief overview of what the authors show, English might be edited with a help of a native English-speaking scientist or a commercially available English proofreader.
Reviewer 3 Report
Comments and Suggestions for Authors
This manuscript presents the role of drinking water disinfection byproducts in initiating cytotoxicity and hyperproliferation in a human lung fibroblast cell line.
The authors used various biochemical assays to assess the antioxidant status and the role of ROS in the induction of oxidative damage in human fibroblast cells after their exposure to different oxidation products (CHCl3, CH2Cl2, and BrCHCl2) and depending on their type and concentration. It has been performed a large number of analyses that examined the relationship between the concentration of oxidant molecules and levels of oxidative stress. The presented results are well described, comprehensively discussed, and supported by statistical analysis.
The methods used are comprehensively presented, the graphical images are of high resolution, the sequence of paragraphs meets the journal's requirements, and the discussion and conclusion are directly related and comprehensively describe the results obtained.
Reviewer 4 Report
Comments and Suggestions for Authors
Dear Authors, the Ms - biomedicines-3117903 can be accepted.
Round 2
Reviewer 1 Report
Comments and Suggestions for Authors
The authors failed to evaluate the relevance of their findings. Just describing the effects without considering the abundant information such as dose-response, different toxic mechanisms, species differences etc of the three halomethanes is of no added value. The effects observed may be non-specific consequences of high concentrations.
ATSDR, EPA, ECHA have presented large documents on the three compounds. Accordingly, liver and kidney are the major targets and metabolism to reactive intermediates is not the major cause of toxicity.
Dichloromethane: Oxidative metabolism to CO. Minor pathway GSH dependent metabolism to the reactive S-(chloromethyl)glutathione. Occupational Exposure Levels bout 180 mg/m3. Major target: CO formation, liver.
Trichloromethane: Formation of reactive intermediates such as phosgene and dichloromethyl radicals which react with cellular components such as fatty acids and phospholipids. The resulting lipid peroxidation accounts at least in part for the hepatotoxic and nephrotoxic effects of chloroform. Occupational Exposure Levels about 2.5 mg/m3. Major targets: kidney, liver.
Bromodichloromehane: The predominant pathway for bromodichloromethane metabolism is cytochrome P450 oxidation and the major route of excretion is expiration of the parent compound or carbon dioxide, smaller amounts of bromodichloromethane are excreted in the urine and feces. Smaller amounts are metabolized via reduction to a dichloromethyl radical or glutathione conjugation catalyzed by glutathione transferase. EPA: air control limit 0.02 mg/m3 based on kidney effects in mice at a NAOEL of 20 mg/m3. Major targets: liver, kidney.
Comments on the Quality of English LanguageNo comments
Reviewer 2 Report
Comments and Suggestions for Authors
The authors have generally responded appropriately to the reviewer's comments.
